# Metal-Printing Defined Thermo-Optic Tunable Sampled Apodized Waveguide Grating Wavelength Filter Based on Low Loss Fluorinated Polymer Material

**Jihou Wang, Changming Chen, Chunxue Wang, Xibin Wang, Yunji Yi, Xiaoqiang Sun, Fei Wang and Daming Zhang ***

State Key Laboratory of Integrated Optoelectronics, College of Electronic Science and Engineering, Jilin University, 2699 Qianjin Street, Changchun 130012, China; wangjh17@mails.jlu.edu.cn (J.W.); chencm@jlu.edu.cn (C.C.); wangcx2019@139.com (C.W.); xibinwang@jlu.edu.cn (X.W.); yiyj@jlu.edu.cn (Y.Y.); sunxq@jlu.edu.cn (X.S.); wang_fei@jlu.edu.cn (F.W.)

**\*** Correspondence: zhangdm@jlu.edu.cn; Tel.: +86-1301-921-6406

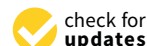

**Featured Application: This paper aims to propose a photonic device which realizes multiple functions of periodic sampling, wide-spectrum filtering, and high side-lobe suppression at the same time. Low loss fluorinated polymer and metal-printing structures are also utilized to reduce optical power propagation loss and reduce driving power consumption, respectively.**

**Abstract:** In this work, thermo-optic (TO) lateral shift apodized sampled waveguide grating for 1550 nm wavelength is designed and fabricated by the metal-printing technique based on fluorinated epoxy-terminated polycarbonates (FBPA-PC EP) and fluorinated epoxy resin (FSU-8) materials. The optical characteristics and thermal stability of the FBPA-PC EP and FSU-8 materials are analyzed. To realize periodic wide-spectrum filtering and suppress the side-lobes of grating, a lateral shift apodized sampled waveguide grating is proposed. The 3 dB bandwidth and wavelength spacing can reach 4.8 nm and 9.7 nm. The side-lobe suppression ratio of proposed device can reach 22.6 dB, which is much better than traditional Bragg grating (6.1 dB). Driving electrical powers of 42.4 mW and 87.2 mW can produce blueshifts of 1.8 nm and 3.5 nm in the measured reflection spectrum, respectively. This device realizes the aim of multiple functions, including periodic filtering, wide-spectrum filtering, and high side-lobe suppression. The device is applicable of realizing signal processing and wavelength division multiplexing (WDM )systems.

**Keywords:** lateral shift apodized grating; fluorinated polymer waveguide; reflect spectrum side-lobes suppression; sampled grating

## 1. Introduction

In modern communication systems, optical communication is a communication method that carries information by light waves. It has the characteristics of large capacity, high speed and low loss [1–5]. At present stage, the improvement of optical communication focuses on wavelength division multiplexing (WDM) technology, miniaturization of devices, and lower optical transmission loss. In a WDM system, the transmission rate can reach 100 Gbit/s by increasing channel density which increases volume of device and the difficulty of realization. A WDM system consisting of sampled gratings has a lower insertion loss, a smaller size, an easier fabrication process, and better dispersion compensation than arrayed waveguide gratings (AWGs) and ring resonators [6–14]. The ideal sampled grating

needs periodic selective filtering wavelength, a box-like response and broad spectrum. However, large side lobes appear when the bandwidth increases. So, the apodized sampled grating is often applied to suppress the side lobes [15–19]. Till now, the most widely used grating has been the amplitude modulated apodized grating, which is usually realized by modulating the ridge height of the grating. The main method employed to fabricate this kind of grating involves an apodized phase mask, apodization speed control with a computer, and control of the emission power to realize ultraviolet (UV) laser writing. The disadvantages of these methods include the requirement of precise processing parameters, complex technology, and only one side of the grating reflection spectrum can be apodized most of time. Therefore, it is necessary to propose a more feasible and effective apodized method for grating. The purpose of this paper is to fabricate a device with a wideband reflection spectrum and larger effect of suppressing side lobes.

In order to realize the sampled apodized grating, waveguide structure is often applied rather than fiber owing to its compact size, more selected materials to fabricate and easily integrated with electronic components on chip [20–24]. At present, several materials can be used for the sampled apodized grating, such as InP, GaAs, silica, and polymer [25–28]. Compared with these inorganic materials, the polymer waveguide can be fabricated with the simplified process of photolithography and wet etching instead of inductive coupled plasma emission spectrometer (ICP) etching, growth technology, diffusion, and laser writing. Further, polymers have flexible structures, easily decorated molecular structure and larger TO coefficients, which can be more suitable to fabricate photonic devices.

In this paper, the metal-printing defined technique is used to realize the sampled apodized waveguide grating based on low loss fluorinated polymer. This device suppresses both side lobes of filtered wavelength by dislocated grating ridges in the form of Gaussian apodization. This device also realizes a wideband reflection spectrum and periodic filtering by sampled grating, which can be applicable for sampling signals. The metal-printing structure is also used in this device to reduce power consumption compare to the structure with upper cladding. The properties of the low-loss fluorinated polymer are introduced in the waveguide material section. The structural parameters of the grating are simulated and optimized. In addition, the comparison of Bragg grating, the traditional apodized grating and our proposed special apodized grating are discussed. Finally, the device is fabricated and the reflection spectrum with TO tuning characteristics is measured. Our proposed device can meet the need of multi-functional integrated circuits WDM system.

## 2. Sampled Apodized Grating Waveguide Structure and Design

### 2.1. Design Principle of Sampled Apodized Waveguide Grating

Our designed sampled grating has a principle that the sampled center wavelength ($\lambda_0$), the 3-dB reflection bandwidth ($\Delta\lambda_b$) and the wavelength spacing ($\Delta\lambda_s$) between adjacent sampled reflection lines are 1550 nm, 5 nm and 10 nm, respectively. Meanwhile, it should have a side-lobe suppression ratio of more than 20 dB. This device has a similar work principle to coarse WDM (CWDM) [29]. The wide bandwidth and wavelength spacing attributes to satisfy the transmission need amongst all end users, and it can also be utilized to realize downstream and upstream traffic. Such a work contributes to keeping the cost of the access network low and economically feasible for subscribers. In addition, $\Delta\lambda_s$ is twice of $\Delta\lambda_b$, which means this device can filter half of the transmission signals in a broad spectrum, and the broad spectrum can be filtered by cascading a sampled grating with a different period. Half of the signal reaches to one area and remaining half signal flows to the other area, which realizes signal separation. This contributes to realizing the signal collection of different usage. Furthermore, $\Delta\lambda_b$ is approximately six times 0.8 nm. It can realize wavelength division multiplexing if narrow band signals are needed. Finally, as the bandwidth increases, the side-lobes of the reflection spectrum also increase. This will extract unwanted transmission signals and increase the crosstalk of device. This work enables the simultaneous implementation of large bandwidth and high selectivity for signals.

## 2.2. Waveguide Material

To realize the sampled apodized grating, low-loss fluorinated photopolymers are used as core waveguide materials. The photopolymers are composed of fluorinated epoxy resin (FSU-8) and fluorinated epoxy-terminated polycarbonates (FBPA-PC EP). The molecular structure of FSU-8 and FBPA-PC EP are shown in Figure 1a and the near-infrared absorption spectrum of the photopolymer film is presented in Figure 1b. Compared with common waveguide materials, the optical absorption loss between 1310 nm and 1550 nm can be reduced by replacing C–H bonds with C–F bonds in the SU-8 material. It can be explained that the atom of hydrogen is replaced with larger atomic mass atom, the resonant frequency decreases and the wavelength of low loss has a blue shift. The refractive index can be changed from 1.495 to 1.565 at 1550-nm wavelength when the content of FSU-8 is changed from 10 mol% to 75 mol%. Table 1 shows the glass transition temperature ($T_g$), the temperature for 5% weight loss of uncross-linked polymers ($Td^C$) and the temperature for 5% weight loss of cross-linked polymers ($Td^D$). The temperature $Td^D$ for 5% weight loss of the cross-linked polymer is obtained to be 303 °C by thermo gravity analysis (TGA). The 25% FSU-8 doped fluorinated photopolymers have better characteristics of low-loss and good thermal stability. The 25% FSU-8 doped photopolymer is adopted in the simulation and experiment.

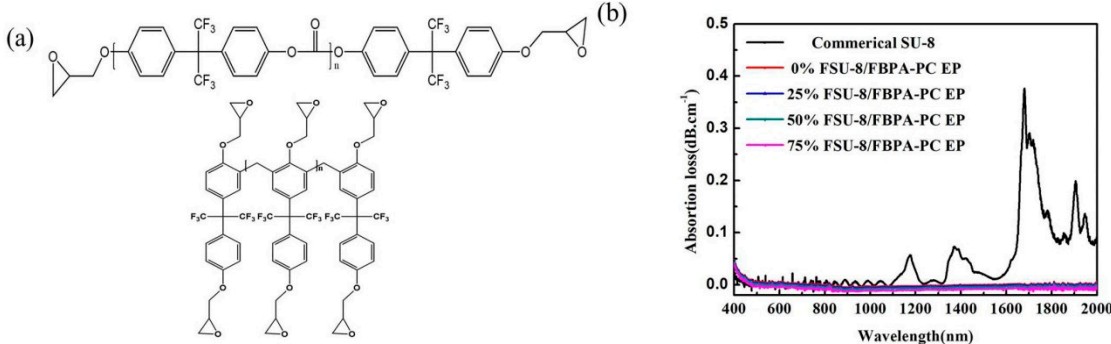

**Figure 1.** (**a**) Molecular structure of fluorinated photopolymer (FBPA-PC EP and FSU-8); (**b**) Near-infrared absorption spectrum of FSU-8 (0%, 25%, 50%, and 75%)/FBPA-PC EP and Commercial SU-8.

**Table 1.** Thermal properties of materials with different content of FSU-8.

| Mixture | FSU-8 Composition (wt. %) | Tg (°C) | Td$^C$ (°C) | Td$^D$ (°C) |
|---|---|---|---|---|
| FSU-8 | 0 | 155.7 | 270.7 | 294.9 |
| | 25 | 158.4 | 277.7 | 302.9 |
| | 50 | 161.9 | 261.4 | 300.3 |
| | 75 | 156.0 | 271.8 | 300.0 |

## 2.3. Optical Modes Analysis

Our designed device mainly realizes wideband reflection spectrum and periodic filtering at 1550 nm wavelength, with mode conversion of $TM_0$ and $TM_1$. The mode conversion is convenient for signal extraction, larger fabrication tolerance and cascading with other photon devices. This paper mainly discusses the function realization of the device, rather than the application of mode conversion. The overall diagrammatic sketch of the sampled apodized waveguide grating is shown in Figure 2a. This sampled apodized waveguide grating is composed of input waveguide, several same sampling periods and output waveguide. Each sampled period consists of an apodized Bragg grating and straight waveguide. The length of sampling period, the grating and duty cycle are defined as $d$, $P$ and $t = P/d$, respectively. The sampled grating contains several parts of the same sampling period, whose number is defined as $N$. The direction of the power transmitting in the waveguide is defined as z axis. The sketch of metal-printing sampled apodized waveguide grating in one sampling period is

shown in Figure 2b. The electrode heater is set to be upon the core FBPA-PC EP. The width of the electrode is designed to be 25 μm and the length of each electrode has the same length of each apodized Bragg grating. Figure 2c shows the diagrammatic sketch view of the grating section. The duty cycle (*Duty* = *W*/Λ) of a Bragg grating is defined as the ratio of length without metal ridge (*W*) to Bragg grating period (Λ). The lateral shifts (Δ*S*) of ridges on both sides are zero at the beginning and end of apodized Bragg grating. The value of Δ*S* is defined as Δ*S₀* at the center of the apodized Bragg grating. The ridge corrugation width (Δ*h*) is also shown in Figure 2c. Δ*h* has important effect on the coupling coefficient and the reflection spectrum. The cross-section profile of this grating is shown in Figure 2d. The 5 μm thickness SiO$_2$ lower cladding lays on the Si bottom substrate, whose thickness is 500 μm. The 2.5 μm thickness fluorinated photopolymer film is applied as the core layer. The sampled grating and straight waveguides are defined by two Al electrodes with 40 nm thickness and most optical power is limited between metals. The effective refractive index of this structure can be calculated by finite difference method (FDM). The refractive indices of the SiO$_2$ bottom cladding layer, fluorinated photopolymer, Al layer, and air are measured as 1.444, 1.526, 1.579 + 11.568 i, and 1.000 at 1550 nm wavelength, respectively. The algorithm of FDM is shown as below.

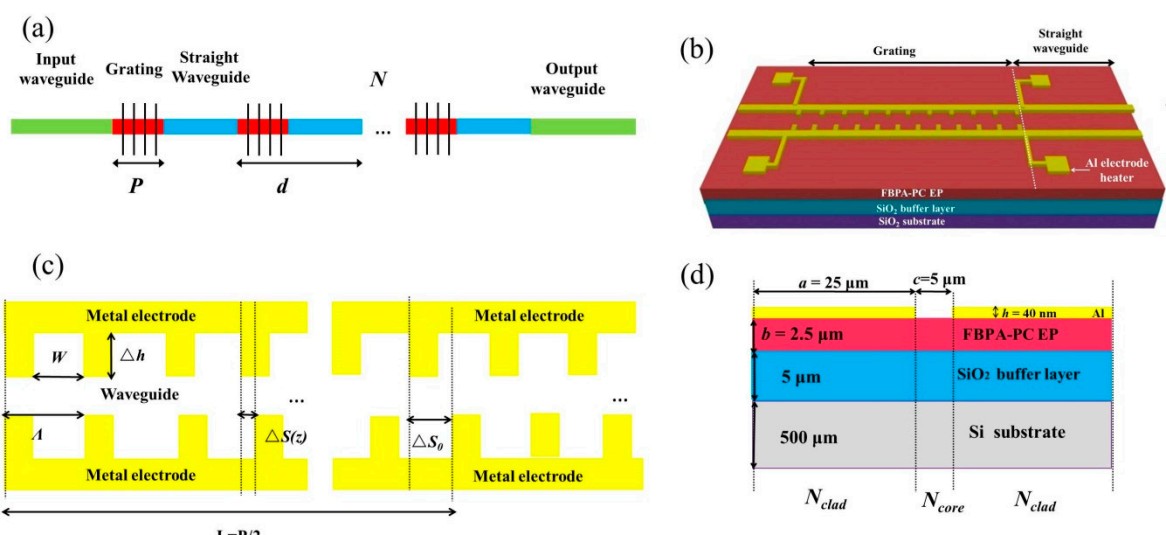

**Figure 2.** (**a**) Overall diagrammatic sketch of the sampled apodized waveguide grating; (**b**) Sketch of metal-printing sampled apodized waveguide grating in one sampling period; (**c**) Diagrammatic sketch view of the grating section with some important parameters; (**d**) Cross-section profile of this grating.

From Maxwell's Equations, the full vector wave equation for the electric field can be obtained as:

$$\nabla_t\left(n^2 E_t\right) + \frac{\partial n^2}{\partial z}E_z + n^2\frac{\partial E_z}{\partial z} = 0 \tag{1}$$

where $E_t$ is the electric field intensity along the x and y directions, $E_z$ is the electric field along $z$ direction, and $n$ is used as the refractive index distribution shown in Figure 2d. Because $n$ is $z$-invariant, $(\partial n^2)/\partial z = 0$, and $E(x,y,z) = E_t\,e^{-j\beta z}$, the vectorial wave Equation (2) can be derived [30].

$$\nabla^2 E_t + \nabla\left(\frac{1}{n^2}\nabla(n^2)\cdot E_t\right) + n^2 k^2 E_t = \beta^2 E_t \tag{2}$$

where $k$ is the wave vector and $\beta$ is the propagation constant. By using the transverse components of $E_x$ and $E_y$, Equation (2) can be expressed as the matrix form of Equation (3).

$$\begin{bmatrix} P_{xx} & P_{xy} \\ P_{yx} & P_{yy} \end{bmatrix}\begin{bmatrix} E_x \\ E_y \end{bmatrix} = \beta^2\begin{bmatrix} E_x \\ E_y \end{bmatrix} \tag{3}$$

It is usual for many optical waveguide devices that the coupling between the two polarizations is weak and negligible. At this time, the semivectorial treatment which refers to TE and TM modes is sufficient. By neglecting the crosscoupling terms $P_{xy}$ and $P_{yx}$, the fullvectorial reduces to Equation (4). The two decoupled semivectorial equations of Equations (5) and (6) are applied to calculate the effective refractive indices of TE and TM modes, respectively [30,31].

$$\begin{bmatrix} P_{xx} & 0 \\ 0 & P_{yy} \end{bmatrix}\begin{bmatrix} E_x \\ E_y \end{bmatrix} = \beta^2 \begin{bmatrix} E_x \\ E_y \end{bmatrix} \tag{4}$$

$$P_{xx}E_x = \frac{\partial}{\partial x}\left[\frac{1}{n^2}\frac{\partial\left(n^2 E_x\right)}{\partial x}\right] + \frac{\partial^2 E_x}{\partial y^2} + n^2 k^2 E_x = \beta_{TE}{}^2 E_x \tag{5}$$

$$P_{yy}E_y = \frac{\partial^2 E_y}{\partial x^2} + \frac{\partial}{\partial y}\left[\frac{1}{n^2}\frac{\partial\left(n^2 E_y\right)}{\partial y}\right] + n^2 k^2 E_y = \beta_{TM}{}^2 E_y \tag{6}$$

In order to solve this Eigen Equation, the waveguide cross section in Figure 2d is discretized into small squares of size $\Delta x \times \Delta y$ (In our calculation, $\Delta x = \Delta y = 10$ nm). The grid being computed is defined as $P$ and the adjoining neighbor points of upper, lower, left and right directions of computed grid $P$ are marked with subscripts $S$, $N$, $E$ and $W$. The refractive indices of these points are defined as $n_S$, $n_N$, $n_E$, $n_W$ and $n_P$, respectively. The expressions of $P_{xx}$ and $P_{yy}$ are listed as Equations (7) and (8).

$$P_{xx}: \begin{bmatrix} 0 & \frac{1}{(\Delta y)^2} & 0 \\ \frac{\alpha_W}{(\Delta x)^2} & n_p^2 k^2 - \frac{2}{(\Delta y)^2} - \frac{2\alpha_P}{(\Delta x)^2} & \frac{\alpha_E}{(\Delta x)^2} \\ 0 & \frac{1}{(\Delta y)^2} & 0 \end{bmatrix}$$
$$\alpha_W = \frac{4(n_W^2 n_P^2 + n_W^2 n_E^2)}{n_P^4 + 2n_E^2 n_P^2 + 2n_W^2 n_P^2 + 3n_E^2 n_W^2}$$
$$\alpha_P = \frac{2(2n_P^4 + n_E^2 n_P^2 + n_W^2 n_P^2)}{n_P^4 + 2n_E^2 n_P^2 + 2n_W^2 n_P^2 + 3n_E^2 n_W^2} \tag{7}$$
$$\alpha_E = \frac{4(n_E^2 n_P^2 + n_W^2 n_E^2)}{n_P^4 + 2n_E^2 n_P^2 + 2n_W^2 n_P^2 + 3n_E^2 n_W^2}$$

$$P_{yy}: \begin{bmatrix} 0 & \frac{y_N}{(\Delta y)^2} & 0 \\ \frac{1}{(\Delta x)^2} & n_p^2 k^2 - \frac{2y_P}{(\Delta y)^2} - \frac{2}{(\Delta x)^2} & \frac{1}{(\Delta x)^2} \\ 0 & \frac{y_S}{(\Delta y)^2} & 0 \end{bmatrix}$$
$$y_S = \frac{4(n_S^2 n_P^2 + n_S^2 n_N^2)}{n_P^4 + 2n_N^2 n_P^2 + 2n_S^2 n_P^2 + 3n_N^2 n_S^2}$$
$$y_P = \frac{2(2n_P^4 + n_N^2 n_P^2 + n_S^2 n_P^2)}{n_P^4 + 2n_N^2 n_P^2 + 2n_S^2 n_P^2 + 3n_N^2 n_S^2} \tag{8}$$
$$y_N = \frac{4(n_N^2 n_P^2 + n_S^2 n_N^2)}{n_P^4 + 2n_N^2 n_P^2 + 2n_S^2 n_P^2 + 3n_N^2 n_S^2}$$

Finally, this Eigen Equation can be solved by Matlab Software. In the calculation, only the maximum eigenvalues are considered. They and their eigenvectors correspond to the modes in the waveguide. The matrix eigenvalue is the square of the mode propagation constant, and the corresponding eigenvectors correspond to the intensity distribution of the mode field.

Figure 3a,b shows the relationships between $N_{clad}$ and $b$ as well as $N_{core}$ and $b$ for TE and TM modes at 1550 nm wavelength, respectively. $N_{clad}$ and $N_{core}$ are defined as the effective refractive indices of the region with and without Al strips, respectively. It can be found that when $b$ is chosen at 2.5 µm, there are only $TE_0$ and $TM_0$ modes in the core waveguide along $Y$ axis. It can also be found the value of $N_{core}$ for $TM_0$ mode in the core and $N_{clad}$ for $TM_1$ mode in the cladding are 1.480 and 1.457, respectively. The value of $N_{core}$ is larger than that of $N_{clad}$. Further, there is $TM_{-1}$ mode in the cladding belongs to the surface plasmon polariton (SPP) mode in the Al cladding structure. The effective refractive index

of $TM_{-1}$ mode in the cladding is larger than that of $TM_0$ mode in the core waveguide, so there is no mode coupling between them. The definition of $W$ is located in the high refractive index region of the grating, which is consistent with the general definition of grating. Figure 3c shows the effective refractive index ($n_{eff}$) of the waveguide related to $c$ when $b$ is 2.5 μm. The value of $c$ is designed as 5 μm to enable $TM_0$ and $TM_1$ modes transmitting in the region between metal strips. The effective indices of $TM_0$ and $TM_1$ optical modes are 1.47243 and 1.46371. The value difference between $TM_0$ and $TE_0$, $TM_1$ and $TE_1$ are smaller than $1.5 \times 10^{-3}$, respectively. The value of $c$ and $b$ are designed as 5 μm and 2.5 μm, respectively. Figure 3d illustrates the cross-section optical field of this structure using the COMSOL software. The optical power in the form of $TM_1$ mode can transport in the waveguide.

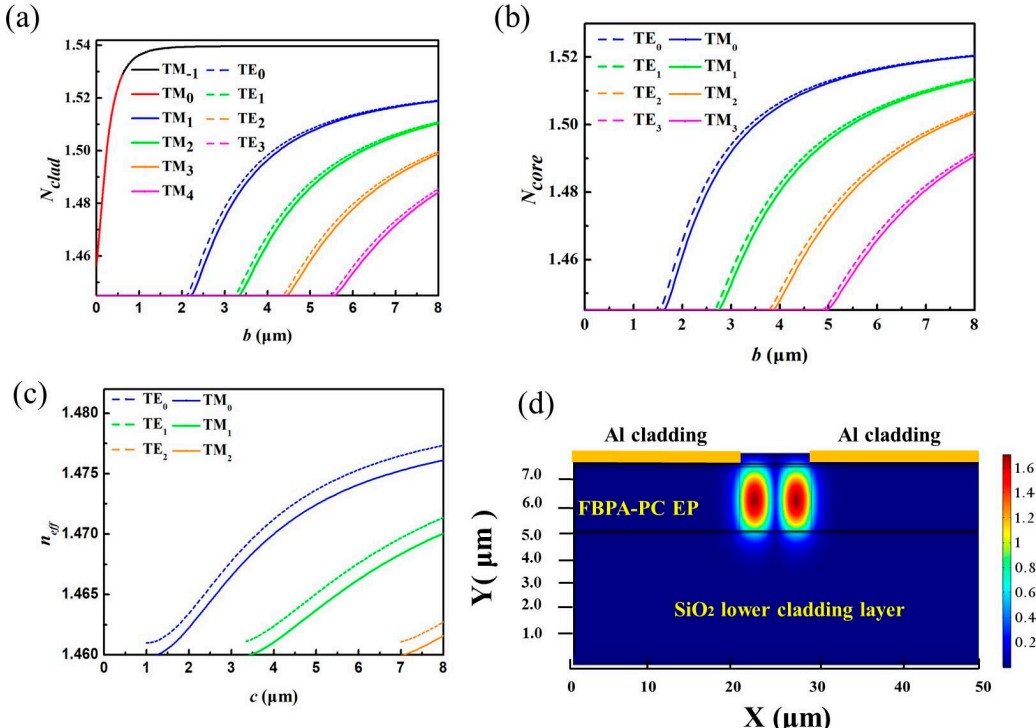

**Figure 3.** (**a**) Relationship between $N_{clad}$ and $b$; (**b**) Relationship between $N_{core}$ and $b$; (**c**) Relationship between $n_{eff}$ and $c$; (**d**) Cross-section optical field of metal-defined waveguide structure.

*2.4. Lateral Shift Apodized Waveguide Grating Design*

In order to realize a sampled apodized grating with the design aim described in Section 2.1, one should first consider a single traditional Bragg grating unit with $TM_0$ and $TM_1$ mode conversion. The value of $\Delta\lambda_b$ is approximate 5.0 nm and SLSR should exceed 20 dB in the single Bragg grating unit. In the traditional mode conversion Bragg grating design, there should be the same value of ridge dislocations $\Delta S_0$ at every grating period to ensure the coupling between $TM_0$ and $TM_1$ optical modes. In general, the value $\Delta S_0$ is generally designed as $\Lambda/2$ to satisfy maximum coupling of $TM_0$ and $TM_1$ optical modes. $\Lambda$ should be satisfied with Bragg phase matching condition, $\lambda_0 = 2 [n_{eff0} \times (1 - Duty) + n_{eff1} \times Duty] \Lambda/m$, where $m$ is the grating order designed as 9, and the fabrication tolerance is larger than that of $m = 1$. For general Bragg grating, the reflectivity is zero when $m$ is even. So odd number of grating order is selected. Further, when $m$ is 9, the minimum fabrication linewidth is obviously larger than when $m$ is 1. In addition, according to the grating equation mentioned above, due to the fabrication error, the offset of resonance wavelength is one ninth of that when $m = 1$. So, the grating order $m$ is selected as 9. In this expression, $n_{eff0}$ and $n_{eff1}$ are the effective refractive indices of $TM_0$ and $TM_1$ modes, respectively.

The transfer matrix method is adopted to solve the single Bragg grating reflection spectrum according to the couple mode equation written in the matrix form. The grating is divided into $M$ uniform sections which are used as 100 in the transfer matrix method calculation. The amplitudes of forward and backward optical mode are expressed as $C_i$ and $D_i$, respectively, where $i$ is between 1 and $M$. In this simulation, $C_0 = C(P) = 1$ and $D_0 = D(P) = 0$. The propagation through each uniform section is described by a matrix defined as $F_i$ in Equations (9) and (10), which is utilized to obtain the reflection spectrum of Bragg grating [32].

$$\begin{bmatrix} C_i \\ D_i \end{bmatrix} = F_i \begin{bmatrix} C_{i-1} \\ D_{i-1} \end{bmatrix} \tag{9}$$

$$
\begin{aligned}
F_{grat} &= F_M F_{M-1} \cdots F_i \cdots F_1 = \begin{bmatrix} S_{11} & S_{12} \\ S_{21} & S_{22} \end{bmatrix} (i = 1, 2, \cdots M) \\
R &= \left| \frac{S_{21}}{S_{11}} \right|^2 \quad F_i = \begin{bmatrix} s_{11} & s_{12} \\ s_{21} & s_{22} \end{bmatrix} \\
s_{11} &= \cosh[s(z_{i+1} - z_i)] - j\frac{\xi}{s}\sinh[s(z_{i+1} - z_i)] \\
s_{12} &= -j\frac{k_0}{s}\sinh[s(z_{i+1} - z_i)] \\
s_{21} &= j\frac{k_0}{s}\sinh[s(z_{i+1} - z_i)] \\
s_{22} &= \cosh[s(z_{i+1} - z_i)] + j\frac{\xi}{s}\sinh[s(z_{i+1} - z_i)] \\
k_0 &= \iint E_0(x,y)\Delta\varepsilon E_1^*(x,y)\,dxdy = \frac{(N_{core}^2 - N_{clad}^2)}{n_{eff0}}\frac{\sin(m\pi\times Duty)}{m}\Gamma_{grating} \\
\xi &= \delta + \sigma_0, s = \sqrt{k_0^2 - \xi^2} \\
\delta &= 2\pi n_{eff}\left(\frac{1}{\lambda} - \frac{1}{\lambda_0}\right), \sigma_0 = \frac{2\pi}{\lambda}\delta_{neff}
\end{aligned}
\tag{10}
$$

In this expression, $z$, $k_0$, $\sigma_0$, and $\Gamma_{grating}$ are the light propagation length, coupling coefficient, refractive index modulation, and power ratio restricted in the region between two metal strips for the traditional mode conversion Bragg grating, respectively. The distance between $z_{i-1}$ and $z_i$ is $P/M$ which indicates every section length is the same. $\lambda_0$ is the resonance wavelength mentioned in the Bragg phase matching condition. Further, $R$ is the calculated reflectivity. In order to reduce the Bragg grating length, $k_0$ should be designed as a large value and it can be adjusted by $\Delta h$. Figure 4a shows the relationship between $\Delta h$ and $k_0$. It can be found that $k_0$ increases faster when $\Delta h$ is smaller than 4.0 µm, and it begins stabilize when the value is larger than 5.5 µm. The selected $\Delta h$ is 5.5 µm in our design and the optimized $k_0$ can reach 200.5 mm$^{-1}$. Figure 4b shows the relationship between the coupling coefficient $k_0$ and $Duty$ when $\Delta h$ is optimized as 4.0 µm. In the traditional design of grating, $Duty$ is often set as 0.5. As $Duty$ is greater than 0.5 and increases, the sine term reduces in the expression of $k_0$. However, the weighted refractive index increases, and a higher power portion is confined in the region between two metal strips, that makes $\Gamma_{grating}$ increase. Thus, the value of $k_0$ depends on the product of sine term and $\Gamma_{grating}$ [25]. The best choice of $Duty$ is 0.8 according to Figure 4b. The optimized $\Lambda$ is 4.8 µm according to the referred Bragg phase matching condition. Figure 4c gives the reflection spectrum at different grating length when $Duty$ and $\Delta h$ are chosen as 0.8 µm and 4.0 µm at the same time. As $P$ increases, the peak reflectivity and $\Delta\lambda_b$ increases. When $P$ is 400.0 µm, $\Delta\lambda_b$ achieves 5.3 nm and it meets our requirements. However, the side-lobe suppression ratio (SLSR) can be calculated as 5.6 dB, which means the side-lobes are large. There is a great need to reduce the side-lobes.

In order to suppress side-lobes and enlarge the SLSR, traditional apodized grating is used to improve the reflection spectrum. In the design of traditional apodized grating, the apodization function is often realized by the height modulation of ridges. The height of ridge is almost zero at the input and output of grating, meanwhile the height has a maximum value at the center of grating. The height of ridge is usually adopted as the Gaussian apodized function, and then the refractive index modulation $\sigma_0$ is realized. The expression of $\sigma_0$ in Equation (10) is related to the location of the grating, which means that $\sigma_0$ should be written as $\sigma(z)$. The expression of $\sigma(z)$ is listed as Equation (11).

$$\sigma(z) = \sigma_0 \exp[-B(z - P/2)^2/P^2] \tag{11}$$

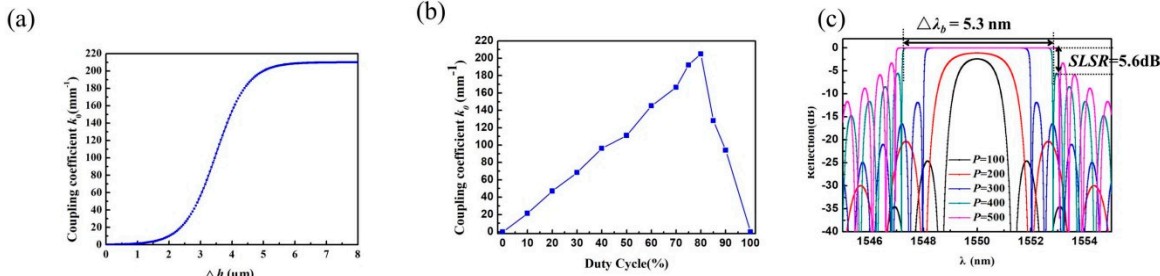

**Figure 4.** (**a**) Relationship between $\Delta h$ and $k_0$; (**b**) Relationship between *Duty* and $k_0$; (**c**) Reflection spectrum at different grating lengths ($P$ = 100, 200, 300, 400 and 500 μm).

In this expression, *B* is usually adopted as 10. For calculating the apodized grating, $\sigma_0$ is replaced by $\sigma(z)$ in Equation (10). With the calculation of Equations (10) and (11), the reflection spectrum of the traditional apodized grating is displayed as Figure 5a. The traditional apodized Bragg grating can improve the SLSR to 22.1 dB on one side of the spectrum and the other side of SLSR hardly decreases (4.1 dB). In addition, fabricating the apodized grating by conventional apodized phase mask and electric arc scanning apodization approach should ensure a precise control of experimental instrument and the fabrication process is complex. Based on the condition, the method of changing $\Delta S$ of ridges is proposed. The $\Delta S$ and coupling coefficient $\kappa(z)$ of the grating are listed as Equation (12).

$$
\begin{aligned}
\Delta S(z) &= \Delta S_0 \exp\left[-B\left(z - \tfrac{P}{2}\right)^2 / P^2\right] \\
\kappa(z) &= \tfrac{1}{2}k_0\left|1 - \exp\left(i\pi \cdot (1 - 2\Delta S/\Lambda)\right)\right| = k_0 \sin\left(\tfrac{\pi\Delta S}{\Lambda}\right)
\end{aligned}
\tag{12}
$$

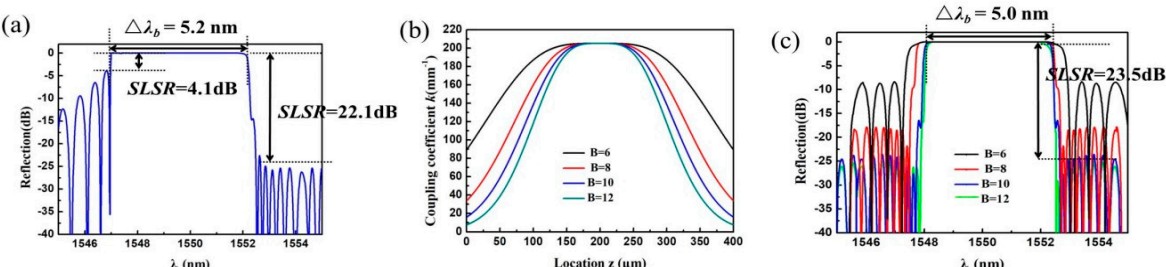

**Figure 5.** (**a**) Reflection spectrum of the traditional apodized grating; (**b**) Relationship of *k* and *B* in our proposed grating; (**c**) Reflection spectrum of the proposed apodized grating ($B$ = 6, 8, 10 and 12).

Figure 5b shows the relationship of $\kappa$ and the location *z* of the grating by calculating with Equation (12). It can be found that $\kappa$ has a larger value of 200.5 mm$^{-1}$ at the center of the grating and exhibits an obvious decrease at both edge sides of the grating. As the apodized coefficient *B* increases, $\kappa$ also decreases at both sides of the grating. When *B* is larger than 10, the change in $\kappa$ is smaller, and the minimum value is stable at approximately 15.0 mm$^{-1}$ at the edge sides of grating. Figure 5c displays the reflection spectrum of the proposed apodized grating according to the combination of Equations (10) and (12) ($B$ = 6, 8, 10 and 12). It can be found that when *B* increases, $\Delta\lambda_b$ has a mild decrease, and SLSR has an obvious improvement. When *B* is chosen as 10, $\Delta\lambda_b$ is calculated as 5.0 nm. The SLSR has a value of 23.5 dB on both sides of the spectrum and shows a nearly ideal filter shape. Compared to the conventional amplitude modulated apodized grating, our proposed apodized grating has the advantages of both side-lobes suppression and simpler fabrication process. This can be explained as in the traditional amplitude apodized grating, the ridge heights of both sides are almost zero and the height gets a maximum value at the center. As the ridge height shows a Gaussian shape, the refractive index of grating also shows in the form of a Gaussian function. The changed value is estimated as 0.005 from the edge to the center of grating. According to the resonance condition of Bragg grating,

the local refractive index in the center of the grating is larger than those of both sides and the resonant wavelength at the center of the Bragg grating is longer than those of both sides. Therefore, both sides of the Bragg grating form a Fabry–Perot cavity and the amplitude apodized grating has a large side lobe in the short wavelength region. As for the proposed lateral shift apodized grating, every grating period has a grating pitch of the same size, which means the effective refractive index is almost the same. The principle of the device is that the coupling coefficient can be changed by the adjusting ridge position. Both side-lobes are suppressed well with this method due to the coupling coefficients on both sides being much smaller than the coupling coefficient at the center.

## 2.5. Apodized Sampled Waveguide Grating Design

In previous section, the design has been optimized well to satisfy $\Delta\lambda_b$ and SLSR in the single Bragg grating. The numbers of parameters of $N$ and $t$ should be optimized well to realize the required $\Delta\lambda_s$. The reflection property of each sampled period is related to the mentioned Bragg grating in the Section 2.4 and connected waveguide with a length of $d-P$. The mentioned Bragg grating depends the bandwidth of the reflection spectrum and the connected waveguide changes the optical phase, which affects $\Delta\lambda_s$. The transfer matrix method is adapted to solve the sampled grating reflection spectrum. In the transfer matrix of sampled waveguide grating, $G_i$ expresses the reflection property in the $i$-th sampled period, which is the product of phase-shift matrix $F_{pi}$ for a phase shift before the mentioned factor of $F_{grat}$ [32]. The expression of $F_{pi}$ is shown in Equation (13) and the calculation of the reflection spectrum is expressed in Equation (14).

$$F_{pi} = \begin{bmatrix} \exp[-j\beta(d-P)] & 0 \\ 0 & \exp[j\beta(d-P)] \end{bmatrix}; G_i = F_{pi}F_{grat} \tag{13}$$

$$G_{grat} = G_N G_{N-1}\cdots G_i \cdots G_1 = \begin{bmatrix} G_{11} & G_{12} \\ G_{21} & G_{22} \end{bmatrix} (i = 1,2\cdots N)$$

$$G_i = \begin{bmatrix} [\cosh(sP)+j\frac{\delta}{s}\sinh(sP)]e^{-j\delta p}e^{j\beta d} & j\frac{\kappa}{s}\sinh(sP)e^{-j\delta[2(i-1)d+P]}e^{j\beta(2i-1)d} \\ -j\frac{\kappa}{s}\sinh(sP)e^{j\delta[2(i-1)d+P]}e^{-j\beta(2i-1)d} & [\cosh(sP)-j\frac{\delta}{s}\sinh(sP)]e^{j\delta P}e^{-j\beta d} \end{bmatrix} \tag{14}$$

$$R = \left|\frac{G_{21}}{G_{11}}\right|^2 \beta = \frac{2\pi}{\lambda}n_{eff0}$$

According to the calculation of Equation (14), the reflection spectrums of the proposed apodized grating are shown in Figure 6a–d when $t$ is selected as 0.5. Figure 6a–d shows the reflection spectra of sampling cycle numbers $N$ of 1, 3, 5 and 10, respectively. As can be seen from these four figures, $\Delta\lambda_s$ is inversely proportional to the sampling cycle numbers $N$, and the reflection spectrum exhibits an envelope. When $N$ is selected as 5, $\Delta\lambda_s$ is 10.0 nm and $\Delta\lambda_b$ is 5.0 nm, which can meet the need of our design.

In the design of the apodized sampled grating, $t$ is an important variable. Figure 7a–d show the reflection spectrum related to $t$ of 0.1, 0.3, 0.5 and 0.7, respectively. As $t$ increases, $\Delta\lambda_b$ and $\Delta\lambda_s$ increase meanwhile. When $t$ is chosen as 0.5, $\Delta\lambda_s$ is 10.0 nm and $\Delta\lambda_b$ is 5.0 nm. So, the optimized $N$ and $t$ are 5 and 0.5, respectively. Figure 8a,b shows the reflection spectrum of traditional sampled grating and the proposed apodized sampled grating when $N = 5$ and $t = 0.5$. It can be found that the SLSR of traditional sampled grating and the proposed apodized sampled grating are 5.6 dB and 23.5 dB, respectively. The ridge dislocation sampled grating can realize the reflection of a wide spectral bandwidth and good effect of side-lobe suppression, which can be useful in the wide spectrum filtering.

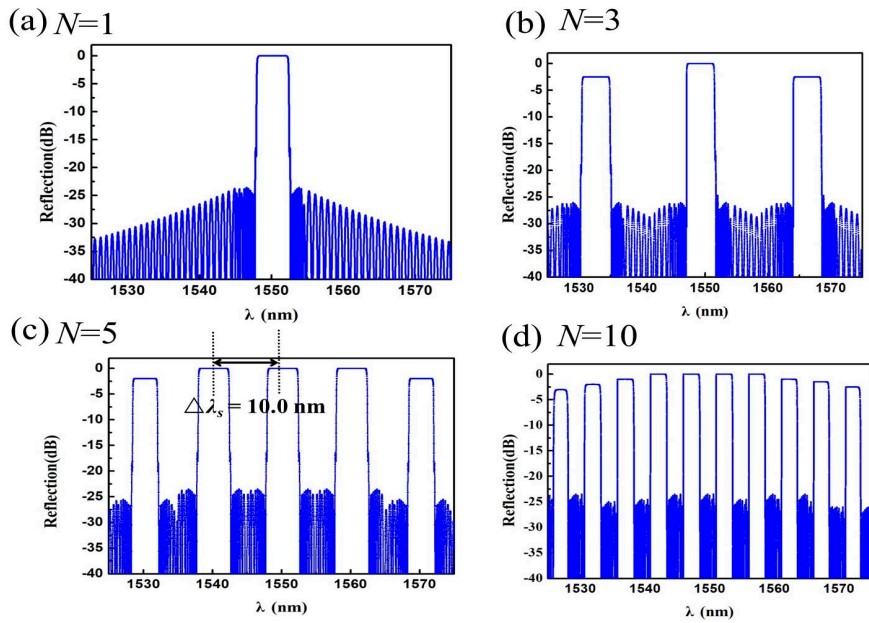

**Figure 6.** Reflection spectrum of the proposed apodized sampled grating related to *N*. (**a**) *N* = 1; (**b**) *N* = 3; (**c**) *N* = 5; (**d**) *N* = 10.

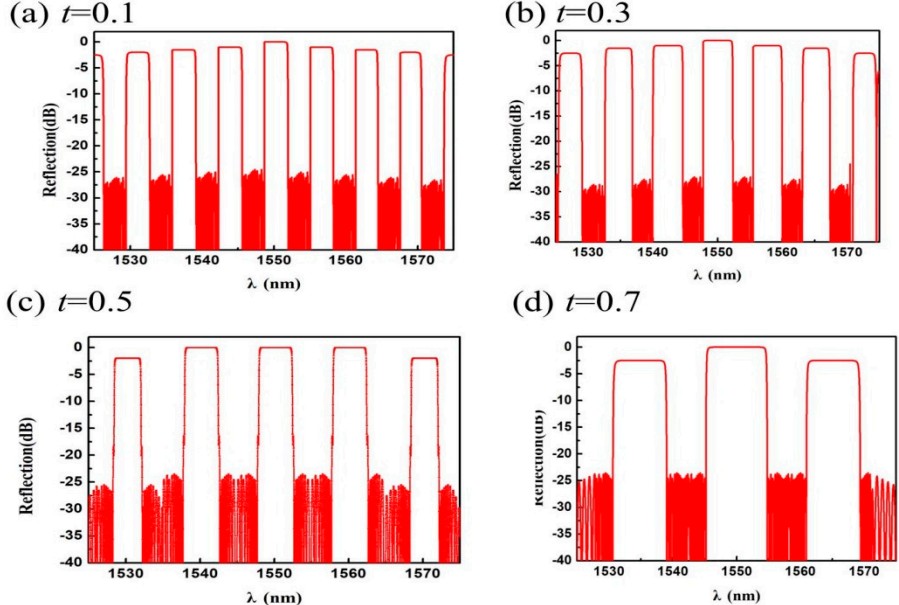

**Figure 7.** Reflection spectrum of the proposed apodized sampled grating related to *t* when *N* = 5. (**a**) *t* = 0.1; (**b**) *t* = 0.3; (**c**) *t* = 0.5; (**d**) *t* = 0.7.

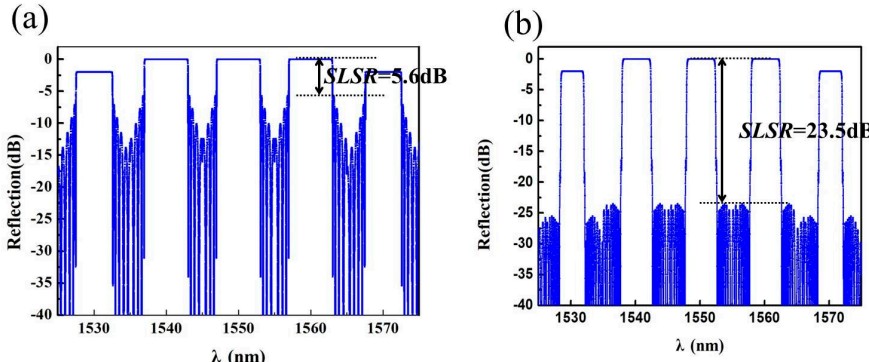

**Figure 8.** Reflection spectrum of (**a**) traditional sampled grating; (**b**) the proposed apodized sampled grating when $N = 5$ and $t = 0.5$.

This metal-printing-defined apodized grating waveguide device can realize a blueshift of the wavelength by applying power to the electrode due to the negative polymer TO coefficient. In order to realize a blueshift of every reflection spectrum peak, probes should be placed on the leftmost and rightmost electrode plates. The value of wavelength blue shift can be expressed as

$$2(\frac{dn}{dT} + \alpha n)\Delta T \Lambda \;=\; m\Delta\lambda \tag{15}$$

In Equation (15), $dn/dT$ is TO coefficient of the material and $\alpha$ is the thermal expansion coefficient of the waveguide material. The TO coefficient and thermal conductivity of FBPA-PC EP are $-1.85 \times 10^{-4}\,°C^{-1}$ and $0.25\,Wm^{-1}°C^{-1}$, respectively. Figure 9a shows the cross-sectional thermal field distribution of the metal-printing heater and Figure 9b shows the reflection spectrum when the temperature change ($\Delta T$) generated from the electrode heater are 0, 10, and 20 °C, respectively. The wavelengths have a shift of 1.9 and 3.7 nm when the values of $\Delta T$ are 10 °C and 20 °C, which are approximately equal to the calculation of Equation (15). This device has the ability to tune the wavelength with its thermo optic effect.

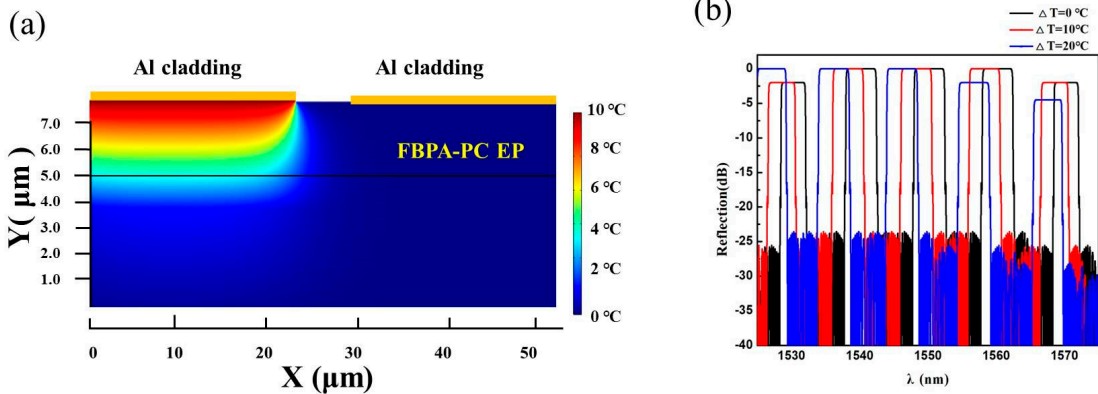

**Figure 9.** (**a**) Cross-sectional thermal field distribution of the metal-printing heater. (**b**) Simulated reflection spectrum of proposed sampled grating when $\Delta T = 0$ °C, 10 °C and 20 °C, respectively.

## 3. Experimental Results and Discussion

The fabrication process for proposed apodized sampled grating is shown in Figure 10. First, a 2.5-µm thin film of 25% FSU-8-doped material FBPA-PC EP was spin coated with the speed of 3000 rpm for 20 s on the SiO₂ lower cladding. Next, a 40 nm-thickness Al film was evaporated on the FBPA-PC EP film with a low vacuum of $5.0 \times 10^{-3}$ Pa. After Al film was evaporated, the positive

photoresist BP212 was spin coated on the Al film and should be baked at 85 °C for 20 min. Then the wafer was exposed to a 200 mW Hg UV lamp power through a positive lithography plate and the exposure time was 4.0 s. Finally, this device was immersed in the concentration of 5‰ NaOH solution for 20 s and flashed by deionized water to remove BP212 photoresist and unexposed Al film. This device was composed of two Al metal strips. Figure 11a shows the top view of the electrode measured by optical microscope (×100) and Figure 11b shows the morphology of apodized sampled grating in one sampling period using optical microscope (×200). Figure 11c,d shows the appearance of the device at the beginning and center position of the grating, respectively. The length of grating in one sample period was 400 μm. The width between two strips could be observed as 5.1 μm and the ridge width was approximately 5.6 μm. The duty cycle could be estimated at 0.8. The lateral shift $\Delta S$ at the beginning and the end of the grating is almost zero in Figure 11c and $\Delta S_0$ is close to $\Lambda/2$ in Figure 11d.

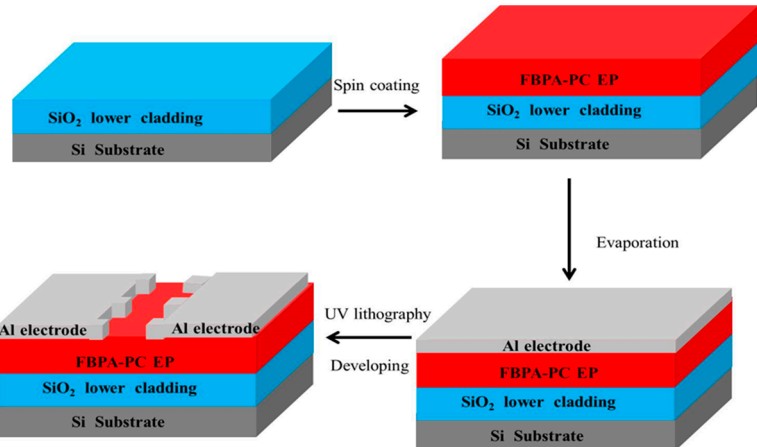

**Figure 10.** Fabrication process for the proposed apodized sampled grating.

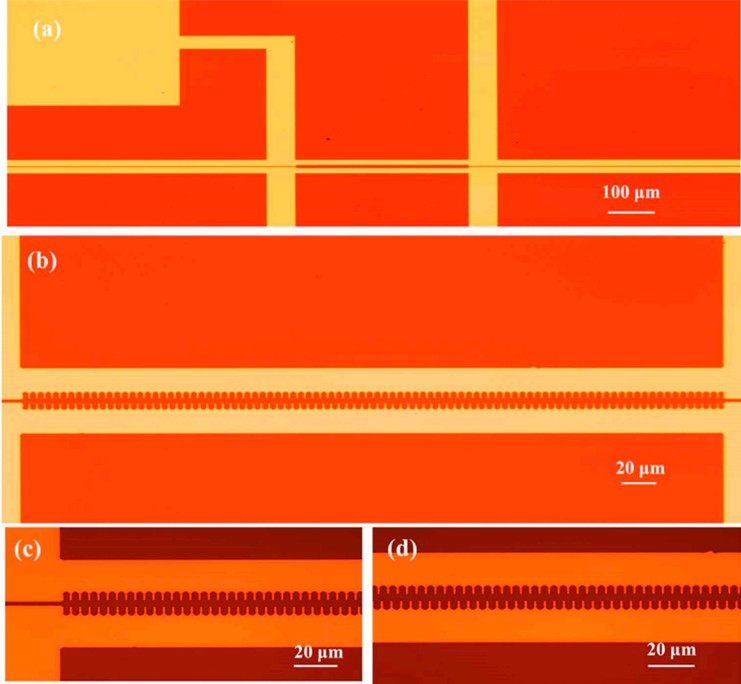

**Figure 11.** (**a**) Top view of the electrode measured by optical microscope (×100); (**b**) morphology of grating region (×200); (**c**) Appearance of the grating at the beginning (×500); (**d**) Appearance of the grating at the center (×500).

Figure 12a presents the schematic diagrams of the measurement system. The apodized sampled tunable waveguide grating was characterized by signal wavelengths in a range from 1530 to 1570 nm by an amplified spontaneous emission (ASE) (SC-5) laser. The electrical modulating power from a signal generator was loaded on the electrodes of the gratings by needle probes. The optical spectrum analyzer (OSA) (MS9740A) was used to detect the reflected power of the grating. The output light signal was divided into two parts by the splitter. One part entered into optical power meter to evaluate the insertion loss and the other part launched into photo detector and oscilloscope (DS4024). The light power entering into the oscilloscope was converted to electrical signals. The propagation loss of the straight waveguide was measured by the cutback method. It was found to be 0.15 dB/cm at 1550-nm wavelength. The insertion loss of the device was directly measured to be about 6.5 dB. Figure 12b shows the signal produced by the signal generator and the signal generated by the broad spectrum ASE laser. The TO switching response of the apodized sampled grating was observed by applying square-wave voltage with amplitude as 1.015 V at a frequency of 300 Hz. The rise and fall switch times were measured as 1.508 ms and 630.5 μs, respectively. The difference between rise and fall switch times can be explained as the fall time mainly depends on the heat dissipation rate. The silica bottom-cladding has a thermal conductivity of 1.2 $Wm^{-1}°C^{-1}$, which is much larger than that of 25% doped FBPA-PC EP material (0.25 $Wm^{-1}°C^{-1}$). The fast heat conduction rate leads to smaller switch fall time. Figure 12c shows the reflection spectrum of traditional Bragg sampled grating. The SLSR was measured about 6.1 dB. The side lobes near each reflection peak were strong. Figure 12d shows reflection spectrums when applying power on the electrode. When there was no heat, the 3-dB bandwidth, wavelength spacing, SLSR were 4.8 nm, 9.7 nm, and 22.6 dB, respectively. When the temperature on the electrode changed 10 °C and 20 °C, the resonance wavelength shifted approximately 1.8 nm and 3.5 nm. Meanwhile, the power consumptions were 42.4 mW and 87.2 mW. It can be calculated the power consumption and wavelength shift has a relationship of 24.91 mW/nm. Table 2 shows the comparison between the proposed polymer apodized waveguide grating and other waveguide gratings. Our device had a larger SLSR for the lateral shift structure to reduce the coupling coefficient at both sides of grating, which is the aim of proposed device. It also had smaller driving power and larger wavelength shift at the same temperature change of the electrode, which could be explained as the electrode heater was directly above the polymer layer and this avoided the upper cladding thermal absorption of traditional waveguide structure containing upper cladding.

**Table 2.** Comparison of other published results for Bragg waveguide gratings.

| SLSR | Wavelength Shift and Driving Power (Temperature) Relation | Reference |
|---|---|---|
| 13.2 dB | / | [18] |
| 10.3–13.8 dB | / | [33] |
| ~12.0 dB | 0.16 nm/°C | [34] |
| ~8.0 dB | 0.082 nm/°C | [35] |
| ~20.0 dB | 37.7 mW/nm | [36] |
| 22.6 dB | 24.91 mW/nm<br>0.175 nm/°C | Present Work |

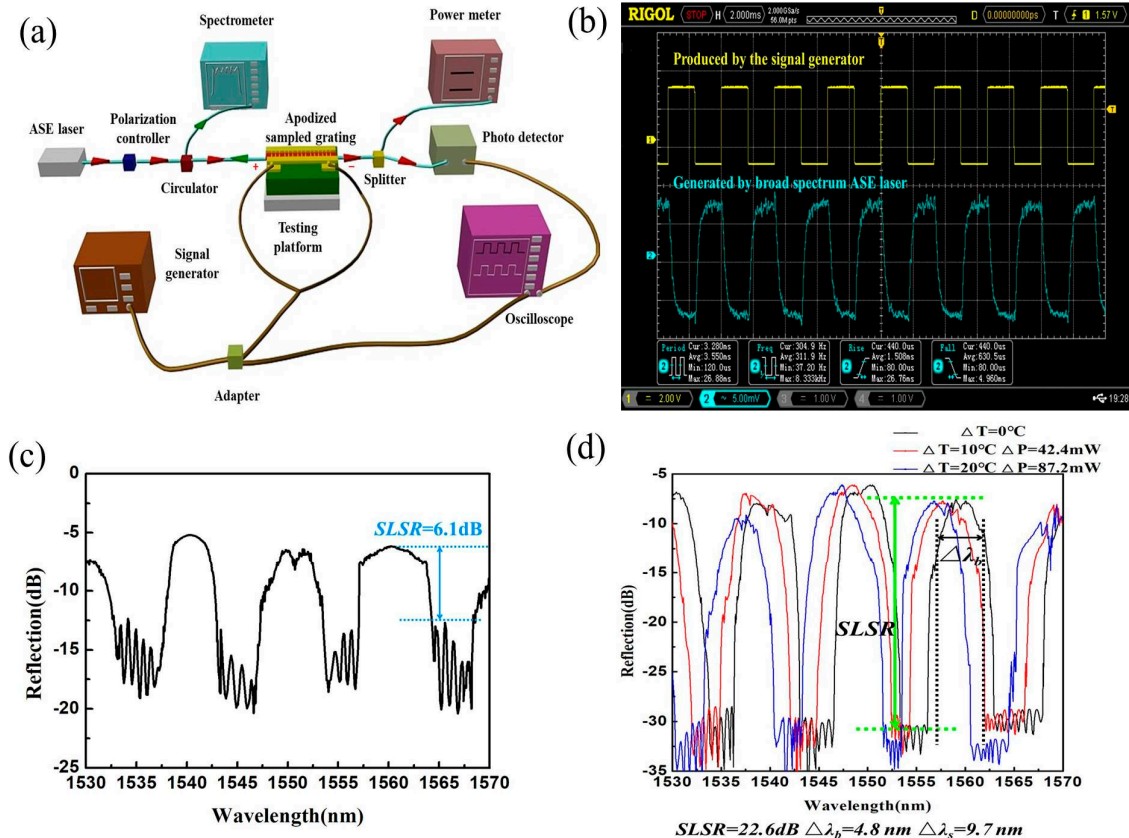

**Figure 12.** (**a**) Schematic diagrams of the measurement system; (**b**) TO switch responses obtained by applying a square-wave voltage at frequency of 300 Hz; (**c**) Reflection spectrum of traditional Bragg sampled grating; (**d**) Reflection spectrum of proposed Bragg sampled grating and its thermo optic tunable characteristic at $\Delta T = 0\ °C$, 10 °C, and 20 °C.

## 4. Conclusions

In summary, a TO lateral shift apodized sampled waveguide grating device based on FBPA-PC EP and FSU-8 was designed and fabricated by the metal-printing technique. This device realizes the effective combination of multiple functions, including periodic filtering, wide-spectrum filtering, and high side-lobe suppression. The 3-dB bandwidth and wavelength spacing can broaden to 4.8 nm and 9.7 nm. This device has a large SLSR due to the location of the ridges with a Gaussian form. The side-lobe suppression ratio can reach 22.6 dB, much larger than that of traditional Bragg grating (6.1 dB), which is the most important aim in this design. In addition, the metal-printing technique can reduce the power consumption for TO tunable devices. The rise and fall response times of this device are about 1.508 ms and 630.5 μs. The applied powers of 42.4 mW and 87.2 mW led to reflection spectrum shifts of 1.8 and 3.5 nm in the test, which can be attributed to the larger TO coefficient and the metal-printing structure. This proposed lateral shift apodized sampled grating will have an extensive application in the WDM system owing to its multiple functions and simpler fabrication process for metal-printing.

**Author Contributions:** Conceptualization, J.W.; methodology, D.Z.; software, J.W.; validation, C.C., C.W. and R.C.; formal analysis, J.W.; investigation, D.Z.; resources, J.W. and D.Z.; data curation, J.W. and Y.Y.; writing—original draft preparation, J.W.; writing—review and editing, J.W., X.Q. and F.W.; visualization, J.W. and Y.Y.; supervision, F.W., C.C. and D.Z.; project administration, X.W. and X.S.; funding acquisition, D.Z. All authors have read and agreed to the published version of the manuscript.

**Funding:** This work was supported by the National Key R&D Program of China (2019YFB2203001), National Natural Science Foundation of China (61875069, 61605057, 61675087) and Science and Technology Development Plan of Jilin Province (20190302010GX).

**Conflicts of Interest:** The authors declare no conflict of interest.

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
