# Peer review of "Metal-Printing Defined Thermo-Optic Tunable Sampled Apodized Waveguide Grating Wavelength Filter Based on Low Loss Fluorinated Polymer Material"

_applsci, doi:10.3390/app10010167_

Round 1
Reviewer 1 Report
The work can be published after considering the following remarks.
On line 49 the abbreviation ICP is not explained.
In line 110 it is not written for which wavelength the refractive index was determined.
In some places (e.g. lines 101 and 102) the delta sign is rotated.
Table 1 uses the Celsius temperature scale, later the Kelvin scale (e.g. line 265) is used. I suggest using the Celsius scale in all work.
Figure 12 a is too small and not legible, please enlarge it.
Reviewer 2 Report
The paper entitled "Metal-printing defined thermo-optic tunable sampled apodized waveguide grating wavelength filter based on low loss fluorinated polymer material" contains two parts:
1. a theoretical / simulation part
2. an experimental part
The main problem with this paper is that it contains a lot of information from simulation to experiment but with not enough explanation per section.
I recommend to enhanced all parts. That will probably make a too big paper, so perhaps separate papers for simulation and experiments should be considered.
The first part involves a lot of electromagnetism theory but it is not enough detailed to appreciate its quality.
Here are some examples:
L115: How is derived equation (1)? This equation seems to be applied to the transverse electric field but what is the shape of n used here (n=n(x,yz))?
L117: "Equation (1) can be approximated as ..." what is approximation used here? At least n=n(x,y) (if no grating) and relation (2) could contain Pxy and Pyx terms. Here x and y components are decoupled, why?
How do you compute TE and TM modes from (1)?
L122: The explanations to obtain relations (3) to (4) are unclear. ==> either it is classical and references should be given, or it is original and it should be detailed.
L167: What is the definition of delta h?
....
I suggest to rewrite the simulation part with the useful details and to extensively use references to refer to basic and advanced methods already known but adapted in this particular simulation.
Reviewer 3 Report
In the present paper, the authors design, fabricate and test a wavelength filter based on a low loss polymer material. The results and experimental procedures are presented in great detail and seem to be original. The applied character of the research is clearly obsereved. Several changes to the manuscript are requested.
1) Please explain all the abbrevations on the Abstract, e.g. FBA-PC, etc. Not everybody will be familiar with them
2) It remained unclear to me what were your design requirements. Please begin your consideration by formulating the design requirements and explaining why these parameters are need. May be introduce a separate Section for that Purpose. By doing so, also the novelty of your manuscript in comparison to the works by other will be better visible.
3) Section 2.2. You calculate the eigenmodes of your waveguides. Why? It seems to me that these results have not been used further. Please explain.
4) Please go through the maniscript in order to improve the style. In general, it is OK, but some passages can be improved, e.g. "The paper aims to have a wideband reflection spectrum and larger effect of suppressing side lobes." --> E.g. The purpose of this paper is to realize ...
Round 2
Reviewer 2 Report
The paper entitled "Metal-printing defined thermo-optic tunable sampled apodized waveguide grating wavelength filter based on low loss fluorinated polymer material" has been substantially improved.
Here are some comments to be answered before publication:
1. L19,abstract: "... blueshifts of 1.8 and 3.5nm in ..." ==> add a space between 3.5 and nm.
2. L151: "... adjoining neighbor points of upper, lower, left and right directions of computed grid P are marked with subscripts S, N, E and W." ==> is it really upper = S, lower = N, left = E and right = W?
3. k^2 is missing in the central term of rel.(5.1).
4. for fig.3a and 3b, what is the value of c?
5. for fig.3c, what is the value of b?
6. Author of Ref.32 is Erdogan T. and not E. Turan.
7. for fig.4a, what is the value of Duty?
8. for fig.4b, what is the value of \Delta h?
9. what are Duty and \Delta h for fig.4c?
10. L234: "The â–³S and coupling coefficient k of the grating ..." ==> k should be k_0.
11. L240: "When B is larger than 10, the change in κ is smaller, and the value is stable at approximately 15.0 mm-1." ==> what do you mean by stable at 15 mmm-1 as the maximum is sill flat at 200.5 mm-1. Better explain that you refer probably to the edges if the grating.
12. what is the value of t in fig.6?
